# Addressing Akrasia in Childhood, Adolescent and Young Adult Cancer Survivors: Implications for Long-Term Follow-Up and Preventive Health Interventions

**DOI:** 10.3390/cancers17081310

**Published:** 2025-04-13

**Authors:** Charlotte Demoor-Goldschmidt, Kristopher Lamore, Zsuzsanna Jakab, Maëlle de Ville de Goyet, Sabine Heinrich, Laura Bathilde, Claire Berger, Laura Beek, Marion Beauchesne, Erika Borszekine Cserhati, Bénédicte Brichard, Louis S. Constine, Jeroen te Dorsthorst, Michele Favreau, Desiree Grabow, Louise Hinckel, Anita Keresztes, Luc Ollivier, Baptiste Sauterey, Roderick Skinner, Eric Thebault, Isabelle Thierry-Chef, Sarolta Trinh, Lorna Zadravec Zaletel, Jelena Roganovic, Marie-Celine Chades-Esnault, Aurore Armand

**Affiliations:** 1Pediatric Oncology-Hematology-Immunology Department, University Hospital of Angers, 49933 Angers, France; louise.hinckel@chu-angers.fr; 2Pediatric Oncology-Hematology-Immunology Department, University Hospital of Caen, 14033 Caen, France; 3Epidemiology of Radiation, U1018 Inserm, Gustave Roussy, 94805 Villejuif, France; 4GCS HUGO, University Hospitals of “Grand Ouest”, 49000 Angers, France; marion.beauchesne@chu-angers.fr; 5Univ. Lille, CNRS, UMR 9193—SCALab—Sciences Cognitives et Sciences Affectives, F 59000 Lille, France; kristopher.lamore@univ-lille.fr; 6National Childhood Oncologic Registry, Hungarian Pediatric Oncology Network, 1085 Budapest, Hungary; dr.jakab.zsuzsa@gmail.com; 7Pediatric Hematology and Oncology Department, Institut Roi Albert II, Cliniques Universitaires Saint-Luc and Institut de Recherche Expérimentale et Clinique, UCLouvain, 1200 Brussels, Belgium; maelle.deville@saintluc.uclouvain.be; 8Les Aguerris Association, 75014 Paris, France; sabine.heinrich@free.fr (S.H.); laurabathilde@gmail.com (L.B.); 9Department of Pediatric Hematology and Oncology, University-Hospital, 42055 Saint-Etienne, France; claire.berger@chu-st-etienne.fr; 10UMR-S1153, Inserm, Paris Descartes University, 75006 Paris, France; 11Department of Psycho-Oncology, Princess Máxima Centre for Paediatric Oncology, 3584 CS Utrecht, The Netherlands; l.r.beek-7@prinsesmaximacentrum.nl; 12Érintettek Association, 1088 Budapest, Hungary; erika.cserhati@gmail.com; 13Pediatric Hematology and Oncology Department, Institut Roi Albert II—Ethics Committee—Cliniques Universitaires Saint-Luc, UCLouvain, 1200 Brussels, Belgium; benedicte.brichard@saintluc.uclouvain.be (B.B.); keresztes_anita@yahoo.com (A.K.); 14Departments of Radiation Oncology and Pediatrics, University of Rochester Wilmot Cancer Institute, Rochester, NY 14642, USA; louis_constine@urmc.rochester.edu; 15PanCare, 1401 AZ Bussum, The Netherlands; jeroen.dorsthorst@pancare.eu; 16University Angers, GRANEM, SFR CONFLUENCES, 49000 Angers, France; michele.favreau@univ-angers.fr; 17Division of Childhood Cancer Epidemiology/German Childhood Cancer Registry, University Medical Center of the Johannes Gutenberg University Mainz, 55131 Mainz, Germany; desiree.grabow@uni-mainz.de; 18Department of Radiation Oncology, ICO, 44800 Saint Herblain, France; luc.ollivier@ico.unicancer.fr; 19Department of Oncology, ICO, 49000 Angers, France; baptiste.sauterey@ico.unicancer.fr; 20Department of Paediatric and Adolescent Haematology/Oncology, Great North Children’s Hospital and Translational and Clinical Research Institute and Centre for Cancer, Newcastle NE7 7DN, UK; roderick.skinner@newcastle.ac.uk; 21Department of Pediatric Oncology, Oscar Lambret Center, 59000 Lille, France; e-thebault@o-lambret.fr; 22Barcelona Institute for Global Health, 08036 Barcelona, Spain; isabelle.thierrychef@isglobal.org; 23Hungarian Pediatric Oncology Network, 1088 Budapest, Hungary; sarolta.trinh@outlook.com; 24Radiotherapy Department, Institute of Oncology Ljubljana, 1000 Ljubljana, Slovenia; lzaletel@onko-i.si; 25Faculty of Medicine, University of Ljubljana, 1000 Ljubljana, Slovenia; 26Department of Pediatric Hematology and Oncology, Children’s Hospital Zagreb, 10000 Zagreb, Croatia; jelena.roganovic@kdb.hr; 27Faculty of Biotechnology and Drug Development, University of Rijeka, 51000 Rijeka, Croatia; 28CAPHI (Centre Atlantique de Philosophie, Atlantic Centre for Philosophy), UR 7463, 44312 Nantes, France; marie-celine.chades@univ-nantes.fr; 29EREPL (Espace de Réflexion Ethique des Pays de la Loire, Ethics Reflection Space of the Pays de la Loire Region), 49000 Angers, France; auarmand@chu-angers.fr; 30Adult Emergency Department, University Hospital of Angers, 49100 Angers, France; 31République des Savoirs—Lettres, Sciences, Philosophie (Republic of Knowledge: Literature, Science, Philosophy)—USR3608—ED540—ENS—PSL, 75005 Paris, France

**Keywords:** survivorship, screening, adhesion, akrasia, patient activation, nudge, shared decision-making, autonomy, tailored information, childhood cancer survivors, young adult cancer survivors

## Abstract

Many survivors of childhood, adolescent, and young adult cancers face challenges in maintaining long-term follow-up care, despite knowing its importance for their health. This phenomenon, known as akrasia, occurs when individuals act against their better judgment and thus engage in behaviors that individuals know are bad for them. Akrasia in this population is influenced by trauma, cognitive impairments, and the desire to prioritize immediate well-being over long-term health. Our study explores why survivors struggle with adherence and proposes solutions that balance autonomy with supportive interventions. By examining ethical principles, survivor experiences, and behavioral insights, we suggest strategies such as shared decision-making, digital tools, and tailored communication to help survivors stay engaged in follow-up care. This research highlights the need for a patient-centered approach that respects individual choices while fostering long-term health and well-being. Our findings contribute to improving survivorship care models, ensuring that they are adaptable, inclusive, and ethically sound.

## 1. Introduction

Childhood, adolescent, and young adult cancer survivors (CAYACS) represent a unique cohort with specific medical, psychological, and social needs. In Europe, over 16,000 persons aged 0–19 are diagnosed with cancer annually. Survival rates have improved significantly over recent decades, with EUROCARE-6 [1] recently reporting a 5-year survival of 81% (95% CI: 81–82) for all childhood cancer cases. Consequently, the number of CAYACS reaching adulthood increases by approximately 8000 each year, bringing the total to nearly 500,000 now living in Europe (https://siope.eu (accessed on 8 April 2025)) [2]. With these improving survival rates, the focus has shifted towards optimizing healthcare and enhancing quality of life (QoL) for survivors, who face a myriad of long-term health challenges. This approach is therefore one of the four pillars of Europe’s Beating Cancer Plan for a stronger European Health Union (https://health.ec.europa.eu/system/files/2022-02/eu_cancer-plan_en_0.pdf (accessed on 8 April 2025)). The ultimate objective of e-QuoL—an EU-funded project—is to address these needs by providing CAYACS with e-health tools designed specifically for them and with their involvement, aiming to help them manage their health.

CAYACS are at an increased risk of developing chronic health conditions throughout their lifetimes. Regular screening and early interventions are crucial to the aim of identifying these conditions and managing risks. Despite the availability of long-term follow-up (LTFU) guidelines and lifestyle recommendations with evidence-based benefits, adherence remains a significant challenge due to missed appointments and non-adherence to medical advice, leading to exacerbation of health risks, both in Europe [3,4] and globally [5,6,7,8]. Some CAYACS suffer from chronic pain [9], fatigue [10], cardiac diseases, or physical limitations due to their past cancer treatments [11]. These factors can lead to a sedentary lifestyle, which further contributes to the risk of developing chronic and serious metabolic disorders such as obesity and diabetes. Additionally, barriers such as limited access to healthy food options or entrenched habits can make dietary improvements challenging. Interventions like adaptive fitness programs and tailored nutrition counseling can help address these issues. Promoting consistent physical activity remains very important, although engagement levels may vary over time due to fluctuating motivation and other obstacles [10,12].

To better understand and address these challenges, an ethics reflection group was formed as part of the e-QuoL project (https://equolproject.eu/ (accessed on 8 April 2025)). This group, comprising multidisciplinary professionals and patient representatives, examined key dimensions that could improve post-cancer care. The discussions led the group to consider the factors influencing adherence to care and the concept of akrasia (i.e., acting against one’s better judgment by engaging in behaviors known to be harmful or counterproductive). For example, some CAYACS may neglect follow-up medical appointments or avoid seeking medical advice despite knowing the long-term risks to their health. Traditionally, akrasia is defined by three core criteria [13]:-Freedom of Decision: The individual must be able to make decisions without cognitive impairments or external constraint.-Intentionality: The action must be deliberate, not accidental or unintentional.-Acting Against Better Judgement: The individual acts contrary to what they judge—after considering all factors—to be the best course of action.

In a previously published article, Chades-Esnault explored the philosophical underpinnings of therapeutic non-compliance, particularly through the lens of akrasia [14]. The author argued that defining akrasia as ‘weakness of the will’ is simplistic and fails to capture its complexity. Akrasia, inherently intentional, goes beyond mere forgetfulness or willpower, involving complex interactions of intention, freedom, and volition. This work suggests that nuanced approaches to patient autonomy—focusing on education, information, and shared goals—can improve patient compliance. By framing therapeutic non-compliance in philosophical terms, this work highlights the complexity of patient behavior and the importance of aligning medical advice with the patient’s values and objectives.

Thus, the aim of this article is to explore the concept of akrasia and its impact on the health behaviors and outcomes of CAYACS. Understanding the role of akrasia could provide insights into why some preventive interventions fail and could facilitate finding strategies to enhance adherence to LTFU recommendations and healthy lifestyle choices.

## 2. Methods

This study is based on an interdisciplinary ethical reflection process conducted within the e-QuoL project (https://equolproject.eu/ (accessed on 8 April 2025)), an EU-funded initiative aimed at improving quality of life for CAYACS. The methodological approach combined a literature review and discussions by the ethics group involving multiple stakeholders [Figure 1].

### 2.1. Litterature Review

A literature review of existing knowledge on adherence behaviors, akrasia, and survivorship care informed the reflection process. While not an exhaustive systematic review, this analysis helped to contextualize survivor behaviors within cognitive, psychological, and systemic models. Insights from previously published empirical studies on survivorship care and patient decision-making were integrated to ensure there was an evidence-based foundation for the ethical discussions.

#### 2.1.1. Identification of Relevant Studies

The literature review began with a structured search of academic databases (Embase, PubMed, PsycINFO) and Google Scholar, using key terms such as “akrasia”, “adherence behavior”, “childhood and adolescent cancer survivors”, “survivorship care”, “long-term follow-up”, and “late adverse effects”. To capture a comprehensive understanding of akrasia in the context of health-related decision-making, relevant studies from medical ethics, psychology, and behavioral sciences were considered. Reference lists of key articles were also examined to identify additional sources.

#### 2.1.2. Study Selection

Studies were included based on their relevance to the themes of akrasia and adherence behaviors among CAYACS. Empirical research, theoretical discussions, and systematic reviews were prioritized. Exclusion criteria included studies focusing solely on adult cancer survivors without discussion of adolescent or young adult populations, as well as articles lacking empirical or theoretical contributions to the concept of akrasia. The selected studies provided a foundation for understanding how akrasia manifests in health behaviors and what factors may influence adherence to LTFU recommendations.

#### 2.1.3. Data Extraction

Data extraction was conducted in two phases by CDG, KL, and MCE. First, titles and abstracts were screened to identify potentially relevant studies. In the second phase, full texts of the selected studies were reviewed in detail to confirm their eligibility. The final list of eligible studies was established based on this review and discussed with the ethical reflection group.

### 2.2. Ethical Reflection Process

A dedicated Ethics Reflection Group was formed, bringing together researchers, clinicians, patient representatives, economists, ethicists, and philosophers (*n* = 27). Members were drawn from the e-QuoL project and also included individuals not involved in e-QuoL. Most of them specialized in CAYACS survivorship, but some worked outside this field, instead bringing additional, separate expertise in other fields. To ensure a survivor-centered perspective, the Ethics Reflection Group included patient advocates and representatives from survivor associations. Their lived experiences provided critical insights into the real-world challenges of LTFU adherence. These perspectives were particularly valuable in discussions focused on the balance between autonomy and beneficence, as well as in those focused on the role of nudging strategies and shared decision-making in fostering adherence.

Local discussions were organized by project partners in each country to explore context-specific ethical issues and to discuss challenges related to LTFU adherence, patient autonomy, and the psychological burden of surveillance, complementing the work of the Ethics Reflection Group. The involvement of local working groups allowed for deeper regional reflections on specific topics such as oncosexuality, shared decision-making, and digital health interventions.

Two major plenary ethical reflection meetings were held (17 May 2024 and 31 October 2024) to address key ethical concerns related to akrasia and survivor adherence. The discussions incorporated both philosophical perspectives on akrasia and practical considerations drawn from survivorship care experiences.

These meeting allowed the participants to share their perspectives regarding the subject discussed and to share their knowledge and experiences. Additional works from the scientific literature were also suggested by the participants as supplements to the articles found in the previous phase of literature review.

### 2.3. Collating, Summarizing, and Reporting the Results

Following the literature review and ethical reflection process, the collected data and insights from the discussions were synthesized to identify recurring themes and key challenges related to akrasia and adherence behaviors among CAYACS. Findings from the literature review were systematically compared with the perspectives gathered from the ethics reflection group to highlight gaps between theoretical models and real-world experiences.

A narrative approach was employed to categorize discussions around major ethical dilemmas and the main questions raised. Hence, the results are not presented as they would be in a systematic review, with numerous details regarding the number of included studies, the participants, and the content. In our analytical approach, we chose to focus on the topics discussed during the meetings of the ethics reflection group, highlighting the main questions raised by the participants in relation to their current experiences, concerns, and perspectives. This approach allowed us to capture the ethical dilemmas and challenges most relevant to the group, ensuring that the discussions were contextualized within the real-world challenges faced by CAYACS in relation to akrasia in LTFU.

### 2.4. Integration into the E-QuoL Project

This ethical reflection is embedded within the broader e-QuoL project, which aims to develop digital health solutions tailored to CAYACS. The study’s findings contribute to the ongoing design of personalized digital tools, decision aids, and communication strategies intended to enhance survivor engagement in LTFU care.

By combining ethical theory, empirical insights, and survivor perspectives, this methodological approach ensures a holistic and ethically sound exploration of akrasia in cancer survivorship care.

## 3. Results

The analysis of the meetings of the ethics reflection group and literature review revealed four main topics: (1) the medical and psychological needs of CAYACS; (2) exploring akrasia in CAYACS: a question never asked; (3) psychological burden of continuous medical surveillance; (4) addressing akrasia and supporting adherence in CAYACS: ethical approaches.

### 3.1. The Medical and Psychological Needs of CAYACS

Compared to the general population, CAYACS bear twice the disease burden by age 45, facing an elevated risk of chronic conditions, including delayed growth, endocrine issues, reduced fertility [15,16,17], and life-threatening diseases like subsequent cancers and cardiovascular complications [18,19,20,21,22,23]. These risks accumulate throughout life without plateauing [24], such that “being cancer-free is not the same as being free of cancer” [25]. There is a growing focus on LTFU care tailored to enhance QoL by preventing or mitigating late treatment effects through health promotion and psychosocial support [26,27]. Cancer survivorship care plans, or “passports,” provide structured follow-up, which is particularly valuable during transitions in care. These plans offer treatment summaries and follow-up recommendations (including healthy lifelong behaviors and screenings) based on global guidelines like those from International Guideline Harmonization Group and PanCare [28,29,30].

However, access to and adherence [31] to LTFU care remain inconsistent across Europe [32,33] and other regions [7,34,35,36], partly due to the failure of these plans to meet survivors’ individual needs [37,38,39]. Psychological barriers, such as anxiety and fear of recurrence [40,41], disrupt long-term engagement. Adherence to long-term follow-up care is not a static behavior; it fluctuates over time and is often influenced by life stages. Adolescents and young adults are particularly prone to periods of non-adherence due to a desire for independence or reluctance to engage with their medical identity. Survivorship care must adapt to these variations, recognizing that patients may re-engage with care at different points in their lives.

Demographic and clinical factors also shape the manifestation of akrasia in CAYACS. For instance, women are less likely to engage in physical activity compared to men, and survivors with higher educational attainment tend to adopt healthier behaviors, such as smoking cessation and regular exercise [10]. These differences highlight the need for personalized approaches with interventions tailored to the survivor’s background and resources.

While adherence is widely recognized as critical to survivor health, it is important to acknowledge that not all patients struggle with this challenge. Many CAYACS display resilience (i.e., ability to cope with and recover in the face of adversity, challenges, or stress), developing new self-management skills to overcome challenges [42,43]. An understanding of this resilience is essential to understanding nuances in adhesion behavior, as it highlights the importance of individual differences in coping mechanisms, motivation, and the capacity to engage with long-term follow-up care. Recognizing and fostering resilience can inform targeted interventions, promoting better health outcomes and enhancing survivorship quality.

Beyond psychological and emotional barriers, survivors face significant practical challenges (e.g., difficulty getting to and from appointments, difficulty getting health or life insurance) and economic challenges (e.g., difficulty returning to work, financial burden) [44,45]. The need to balance follow-up care with professional responsibilities, family commitments, and personal priorities often relegates medical appointments to a secondary priority. Access to specialized care remains uneven, particularly in underserved or rural areas, where survivors must navigate long travel distances and a lack of local resources [10]. Moreover, the financial burden of long-term follow-up care extends beyond direct medical costs to include transportation expenses, lost income due to missed work, and administrative fees. Survivors from lower socio-economic backgrounds face disproportionately greater challenges, as these accumulated costs create a substantial barrier to adherence. Discussions within the ethics reflection group highlighted the complexity of these barriers, particularly in systems where follow-up care is not financially incentivized or sufficiently funded. Participants underscored that perceptions of cost—whether actual or symbolic—may dissuade survivors from seeking care, especially after years of free treatment during active cancer therapy. This economic dynamic often leaves survivors questioning the necessity of follow-ups, further eroding adherence. Of note, individuals from lower socioeconomic groups often face pressing concerns (e.g., employment challenges, housing instability, family responsibilities) that overshadow their ability to prioritize their health [46]. Addressing these barriers requires systemic reforms, including mechanisms for providing financial support, regional adaptations, and stronger communication about the long-term benefits of follow-up care.

Additionally, the psychological models of Self-Determination Theory (SDT) and the Health Belief Model (HBM) provide valuable insight into survivors’ struggles with adherence. SDT posits that behavior change is more sustainable when individuals are autonomously motivated, emphasizing the importance of supporting basic psychological needs such as autonomy, competence, and relatedness [47]. However, cancer survivors may experience diminished intrinsic motivation due to prolonged reliance on external controls during treatment [48,49]. The HBM provides a complementary perspective, explaining how perceptions of susceptibility, severity, benefits, and barriers impact health behaviors [50]. Survivors’ emotional fatigue and their desire to distance themselves from a medical identity can distort perceptions of the benefits of adherence to long-term supportive care. These emotional and cognitive factors can lead to avoidance behaviors, making it harder for survivors to prioritize follow-up care. These models emphasize the need to address both psychological and systemic factors to improve survivors’ long-term engagement in follow-up care.

### 3.2. Exploring Akrasia in CAYACS: A Question Never Asked

A PubMed search using the keyword “akrasia” retrieved 24 articles published between 1976 and 2024, none of which were linked to cancer. This highlights a gap in knowledge about the concept of akrasia within the context of CAYACS.

Although no studies to date have directly compared akrasia in CAYACS to akrasia in healthy populations, the psychosocial and medical challenges faced by survivors—including trauma, disrupted development, and complex follow-up trajectories—suggest a distinctive context in which akratic behaviors may manifest differently.

After treatment, CAYACS often experience emotional ambivalence, a mix of relief with anxiety and uncertainty [51,52,53]. The psychological toll of surviving a life-threatening illness often lingers long after treatment [54], potentially manifesting as post-traumatic stress disorder (PTSD) [15,55] or avoidance behaviors, such as skipping follow-up appointments to avoid triggering distressing memories. Studies indicate that survivors face psychosocial issues such as disrupted schooling, unemployment, and social isolation, which can potentially create long-term health management challenges [56,57,58,59]. The St. Jude Lifetime Cohort Study highlights that these psychological effects may reduce patients’ willingness and ability to manage their health and engage in preventive behaviors independently [60]. These external psychological constraints compromise freedom of decision, making us doubt whether these behaviors can really be considered akratic in the strict sense; rather, they may fit only a broad definition of akrasia.

Survival of a life-threatening illness can contribute to certain forms of akrasia that arise from an inability to project oneself into a distant future—a trait strongly influenced by traumatic psychological experiences. For instance, survivors may experience a “temporal decoupling” effect, in which immediate actions conflict with long-term health intentions. This shift in values may be intensified by the psychological impact of a life-threatening illness, encouraging present-focused behaviors over future health considerations. This disconnect may be amplified by a “superhero” mindset in some survivors, a sense of invincibility in which the desire to savor the present moment overshadows long-term health considerations. This inclination to prioritize the immediate—whether for joy, freedom, or defiance—can lead to a conscious decision to disregard medical advice. Thus, the survivor’s will may oscillate between rational health decisions and a drive to find pleasure and meaning in the present. In these cases, akratic behavior may be not a failure of will [61], but a fleeting shift in preferences that changes the survivor’s planned actions.

Normative medical expectations may prompt survivors to outwardly conform to recommendations, presenting adherence as their ‘best judgement’. However, this adherence may reveal itself to be driven more by societal or medical expectations than by genuine personal commitment. This idea is derived from Richard Hare’s theory of moral judgment ‘in inverted commas’ [62]. Such fragile adherence aligns superficially with medical advice but lacks the stability of intrinsic motivation, making it prone to disruption. This dynamic underscores the complexity of akrasia; adherence appears genuine but is often influenced by unresolved internal conflicts and external pressures rather than a true, sustained commitment.

Cancer treatment may interfere with educational and career progression [63,64], affecting social reintegration and resulting in lower educational achievement and higher unemployment rates, with both educational attainment and employment being critical indicators of QoL [65,66,67,68,69]. Additionally, chemotherapy and brain irradiation can cause chronic fatigue [70] and cognitive impairments [71,72,73] that affect decision-making and executive functioning. Studies in adults show that chemotherapy-related cognitive impairment significantly correlates with reduced self-care ability and reduced QoL, underscoring how cognitive limitations hinder self-management [74]. This suggests that a continuum likely exists between akrasia and nonakrasia, depending on how much cognitive or psychological impairments impact decision-making [60,75,76].

All these points raise a pressing question about accountability in healthcare: when illness impairs decision-making, to what extent are patients responsible for actions that oppose their better judgement? In such contexts, true autonomy is compromised.

Adolescents and young adults are especially vulnerable, as their diagnosis and treatment coincide with critical developmental milestones like identity formation and independence. Interruptions in normal development can negatively impact physical well-being (energy, sexuality, fertility, body image), psychological well-being (due to fear of recurrence and feelings of difference) [76,77], health-related QoL, and healthy behaviors [52,60]. This dynamic echoes the internal conflict of akrasia, where survivors may outwardly adhere to medical advice but struggle to internalize the advice and act consistently. Addressing these psychological barriers is essential to crafting interventions that improve long-term health outcomes and prevent disengagement.

### 3.3. Psychological Burden of Continuous Medical Surveillance

The psychological burden of continuous medical surveillance, often termed ’scanxiety,’ [78,79,80] significantly impacts CAYACS, evoking fear, anxiety, and emotional exhaustion as routine follow-ups serve as reminders of vulnerability and potential relapse. For some survivors, these experiences may exacerbate PTSD symptoms, prompting avoidance behaviors that undermine adherence. Understanding the psychological burden of continuous medical surveillance reveals how seemingly minor logistical challenges, such as scheduling difficulties or administrative barriers, can amplify these avoidance tendencies. Simplifying follow-up processes (through, e.g., telemedicine options, patient-centered communication, and flexible scheduling…) and providing consistent support are practical ways to reduce distress and improve adherence to follow-up, overcoming these obstacles.

As noted by Lutz [81], patient autonomy in chronic care operates as a negotiated order, where tensions arise between healthcare providers’ normative expectations and patients’ actual capacities. For CAYACS, this dynamic underscores the importance of care strategies that balance empowerment with realistic support, particularly given the psychological and cognitive barriers many survivors face.

The emotional burden of surveillance also raises ethical questions about balancing beneficence with respect for autonomy. This ethical dilemma becomes more acute when considering the limits that should be respected when recalling patients who miss follow-up appointments. While regular follow-up is crucial for early intervention, the psychological toll challenges the principle of nonmaleficence. Providers must balance preventing harm through engagement with respecting a patient’s decision to disengage.

A previously published ethics reflection on CAYACS survivorship suggests several strategies to address this tension [40,41]:Personalized communication [39,82,83,84]: Tailoring follow-up invitations to reduce the perceived burden and emphasize patient autonomy in making the decision.Shared decision-making [26,85,86,87,88,89]: Engaging patients from the outset to align follow-up plans with their values and preferences, potentially reducing psychological strain and improving adherence.Flexible follow-up options [3,90,91,92]: Offering alternatives to traditional in-person visits, such as telemedicine, that may mitigate distress associated with hospital visits, fostering engagement and respect for autonomy.Better collaboration and communication [39,83,84,85,86,93,94,95,96,97] between all the professionals on the patient’s healthcare team

These strategies constitute a patient-centered approach that carefully balances beneficence, nonmaleficence, and autonomy, enabling providers to support CAYACS through the complex demands of ongoing surveillance.

Moreover, some CAYACS might engage in risky behaviors, such as substance use, reckless driving, or neglecting follow-up care, as a way to reclaim a sense of normality and independence. These actions can often be understood as attempts to regain autonomy or to challenge the medicalized identity imposed during their illness. By rejecting follow-ups or engaging in rebellion-like behaviors, survivors may seek to assert independence from a past defined by dependency and restrictions. Such behaviors may also stem from a desire to distance themselves from their medical experiences or to reassert control over their lives and bodies after enduring months or years of constraints imposed by medical treatments and appointments. While these behaviors may temporarily alleviate feelings of being ‘different,’ they often carry significant long-term health risks that can undermine survivorship. In this context, akrasia may arise from a desire to appear or feel as normal as possible, leading individuals to prioritize short-term comfort or social conformity over long-term health considerations. To counteract these tendencies, care strategies should empower CAYACS through shared decision-making and create opportunities for them to reclaim autonomy in healthier ways, such as through peer-led programs or co-created survivorship plans.

### 3.4. Addressing Akrasia and Supporting Adherence in CAYACS: Ethical Approaches

For many CAYACS, adherence to health recommendations involves overcoming complex psychological and motivational barriers, rather than a mere lack of knowledge or intent. This can be understood through the lens of control, engagement in health decisions, and patient activation. Patient activation is defined as a patient’s ‘willingness and ability to take independent actions to manage their health’ [98], and it is a key element in adherence to health behaviors [99]. Highly activated patients are less likely to exhibit akratic behaviors since they possess the motivation and confidence to manage their health proactively. While high activation correlates with improved health outcomes and adherence [100], low activation, especially when coupled with akrasia, can lead to superficial adherence. This weak adherence often arises from external pressures rather than personal commitment, making long-term follow-up difficult. The St. Jude Lifetime Cohort highlights that CAYACS with lower activation levels are more likely to engage in suboptimal health behaviors such as reduced physical activity and poor diet, although both physical activity and diet are critical for preventing long-term health complications [60]. Different interventions targeting activation, including structured education and psychological support, could help bridge this gap and promote sustained adherence. Certain facilitators can promote engagement. Positive influences such as perceived health benefits, professional support, and peer encouragement have been shown to help survivors adopt healthier behaviors. For instance, survivors who report feeling physically energized or supported by family and friends are more likely to maintain behaviors like regular physical activity or dietary changes [10]. Including these elements in survivorship care plans could mitigate akrasia by fostering intrinsic motivation.

Nudge theory [101,102,103] and shared decision-making [104,105,106] provide interesting possibilities for guiding patients toward long-term health goals. As Thaler and Sunstein [107] proposed, nudge theory involves designing subtle environmental or choice-based interventions to guide individuals toward beneficial behaviors while preserving their autonomy. For example, small adjustments to choice provision—such as default options, simplified decision pathways, or timely reminders—can significantly influence behaviors while preserving autonomy, making such adjustments a powerful tool for enhancing follow-up adherence and promoting long-term lifestyle changes in survivors. For CAYACS, nudges might include personalized reminders, simplified care procedures, or digital health tools that reduce cognitive and emotional burdens contributing to akrasia. While these interventions are designed to help patients in acting in their own best interest, there is a risk they might infringe upon their ability to make fully autonomous decisions. This raises ethical concerns about autonomy, as they could influence patients to make choices they aren’t fully committed to. To address these concerns, nudges should prioritize transparency and focus on enabling informed decision-making without coercion. Shared decision-making complements nudges by directly involving patients in creating care plans that align with their values, fostering commitment to follow-through.

Shared decision-making complements nudges by involving patients in creating their care plans, aligning care with their values and increasing commitment to follow-through. This approach enables healthcare providers and patients to collaboratively shape care plans, respecting individual preferences and addressing emotional and psychological barriers to adherence. As explored in recent studies, interactive digital tools offer a practical way to integrate shared decision-making, fostering autonomy while enhancing adherence [108,109,110,111]. For example, decision aids can simplify complex medical decisions, helping survivors weigh options against their long-term goals and personal values. By focusing on patient engagement in decision-making, healthcare providers can help patients balance their “best interest” from a medical perspective with their immediate emotional well-being, fostering a sense of ownership over the care process.

This also raises questions about a patient’s “best interest”: should the focus be on the medical community’s long-term health priorities or on the patient’s perspective, which might value immediate emotional well-being? Negotiation and shared decision-making play a critical role here, allowing healthcare providers and patients to co-create care plans that respect patient autonomy while addressing barriers contributing to akrasia. This patient-centered dialogue fosters a commitment to the care plan, engaging CAYACS in a discussion about their preferences, values, and concerns to empower them and support adherence. Yet, ethical questions remain—how can we ensure that negotiation respects the patient’s authentic will and avoids subtle reinforcement of medical authority?

Patient activation is the key in this balance. Encouraging patient activation through structured education and personalized interventions can mitigate the impact of akrasia. Highly activated patients are more likely to reconcile medical advice with personal values, balancing long-term health goals with short-term needs and thus preserving autonomy and empowering informed decision-making that reflects long-term best interests. For less-activated patients, autonomy questions become more complex. Should professionals push harder for adherence or respect the patient’s subjective sense of their best interest, even if their consequent choice diverges from medical recommendations?

Healthcare professionals are invited to reflect on these tensions and challenge adherence dogmas. Rather than assuming that non-adherence stems from a lack of will or discipline, it may be essential to consider whether patients’ actions align with their deeply felt values, even if those prioritize short-term fulfillment over long-term outcomes. Thus, nudges should be seen as transparent and supportive tools, not coercive or manipulative instruments. Some authors refer to the ’Ulysses contract’ concept to illustrate how temporary coercion, requested by an individual, can be used to help them achieve their goals [112]. This idea draws inspiration from the myth of Ulysses, who, in order to hear the sirens’ song (which famously drove sailors mad, causing them to throw themselves into the sea), asked his crew to tie him to the mast of his ship to prevent him from acting on impulse. In medicine, this can be translated into a request from the patient themself for temporary restraint that will help them achieve goals and could be a type of response to the problem of akrasia. This question raises a number of ethical issues, such as the legitimacy of limiting respect for individual freedom by allowing patients to submit, even voluntarily, to a device that partially and temporarily deprives them of their freedom.

## 4. Discussion

### 4.1. Clinical Implications: Effective Interventions for Addressing Akrasia and Health-Policy Implications

Recognizing specific drivers of non-adherence—such as trauma, PTSD, cognitive limitations, or mistrust—could enable healthcare providers to refine and tailor their approaches [Table 1]. While akrasia has not been extensively studied in this population, certain hypothetical interventions may provide a basis for exploring strategies that could support adherence more effectively. Psychological strategies such as cognitive–behavioral therapy and motivational interviewing could help survivors strengthen their commitment to health-promoting behaviors by fostering intrinsic motivation and addressing ambivalence. Social support from family, peers, and healthcare professionals may also play a critical role, as family members can reinforce positive health habits and peer support groups provide a community that supports behavior changes and reduces feelings of isolation [113]. Conversely, for patients whose actions are primarily driven by cognitive impairments or psychological trauma, the focus should be on psychological support, stress reduction, and more structured and supported follow-up. It is also important to consider that akrasia may extend beyond the patient to their family, particularly to parents or caregivers. Parents, faced with the emotional and logistical challenges of managing their child’s care, might experience similar struggles in adhering to follow-up schedules or promoting recommended behaviors, especially when burdened by fear, trauma, or conflicting priorities. Interventions aimed at empowering the family unit, offering psychoeducation, and fostering parental self-efficacy could play a key role in addressing these dynamics and improving overall adherence.

To address the challenges of re-engagement, healthcare providers should foster a welcoming and non-judgmental environment. Survivors who have experienced periods of non-adherence often feel guilty or fearful that returning to care will result in negative consequences, such as the discovery of advanced complications. Clear and compassionate communication, emphasizing that all patients are welcome regardless of their past adherence, can encourage them to return to care. Furthermore, framing follow-ups as opportunities for proactive health management—rather than as moments for assessment of past behaviors—can be a powerful tool in rebuilding trust. Survivors should be encouraged to view these visits as part of an ongoing partnership in their health in which their participation is valued and their experiences are acknowledged. It is important to focus on the positive aspects of re-engagement, such as the ability to monitor health, prevent future complications, and improve overall quality of life. By adopting this approach, healthcare providers can help to reduce feelings of shame or fear, making it easier for survivors to re-establish trust and feel motivated to engage in long-term care.

Tools co-constructed with survivors, such as digital health applications or structured follow-up plans, may offer a personalized approach that fosters a sense of ownership and aligns with survivors’ needs and values. Digital tools have shown promise in empowering cancer survivors by addressing critical barriers to engagement and adherence. According to Groen et al. [114], five key components of empowerment exist: autonomy and respect, knowledge, psychosocial/behavioral skills, community support, and perceived usefulness. Digital tools primarily support the first three by providing tailored educational content, interactive guidance, and actionable recommendations, enabling patients to better understand and manage their follow-up care [108,109,115,116,117,118]. For childhood cancer survivors facing cognitive challenges, integrating such tools with adaptive features like reminders, simplified instructions, and interactive educational content could address barriers related to cognitive deficits while reinforcing empowerment. Although the quantitative impact on cognitively challenged populations remains underexplored, the broader evidence highlights their potential for improving knowledge, adherence, and QoL. As Groen et al. emphasize, IT-based solutions align with key components of empowerment, such as autonomy and respect, knowledge acquisition, and psychosocial skills development, suggesting that they are suitable for use in tailored interventions in this vulnerable group [114].

Nudge theory and shared decision-making can bridge the gap between knowledge and activation and that between activation and action by personalizing care to fit the patients’ values, thus enhancing activation and reducing adherence barriers. As demonstrated in recent studies [119], the inclusion of electronic patient-reported outcome measures (ePROs) in the design of shared-decision-making frameworks may further empower patients by individualizing follow-up care and enhancing satisfaction without increasing provider burden. Using ePROs to tailor follow-up care can reduce the use of unnecessary in-person consultations without compromising patient satisfaction or treatment adherence [120]. Such approaches could offer cost-effective solutions to managing the long-term care of childhood cancer survivors while ensuring that their unique needs are met effectively. For CAYACS, these tools could play a critical role in tailoring follow-up care to their specific needs, particularly for those with cognitive challenges. By providing real-time feedback, simplified interfaces, and actionable insights, ePROs empower patients to actively participate in their care while easing the logistical and emotional burdens of follow-up. Decision aids, including brochures, online tools, and structured care plans, offer practical support by breaking down complex medical decisions into manageable steps.

Balancing these approaches helps to mitigate akrasia while respecting the survivors’ autonomy and psychological state. Moser et al. [121] emphasized that the dual dimensions of autonomy—freedom of action (negative autonomy) and the capacity to act according to one’s values (positive autonomy)—are critical for patient-centered care. Nurse-led shared-care models can support this nuanced autonomy by fostering continuous interaction and understanding between patients and providers. Empowering survivors is a cornerstone of effective follow-up care, as empowerment fosters stronger adherence and engagement [108]. Jørgensen et al. [113] identified key enablers of patient empowerment, such as active participation, supportive interactions with healthcare providers, and peer support networks. For CAYACS, incorporating these elements into care plans could enhance their sense of control and reduce the psychological burden of adherence, ultimately aligning care with their personal values and improving outcomes.

Ensuring equitable access to comprehensive survivorship care for all CAYACS, including those with potentially akratic behaviors, will require policy support and resource allocation. Policymakers should prioritize funding for integrated, multidisciplinary lifelong-care programs that incorporate psychological support, social services, and digital health solutions. Economic models should recognize the long-term benefits of socio-professional integration and emphasize the value of preventive medicine over curative approaches. These tools can improve accessibility, provide personalized support, and enhance overall quality of care for survivors. Standardized programs for transition from pediatric to adult care that include clear guidelines, coordinated communication between providers, and education for survivors and their families about the importance of follow-up care, alongside training for healthcare providers in addressing a broad range of non-adherence factors, may be instrumental in providing continuity of care. Providers should be equipped with skills in motivational interviewing, cognitive–behavioral techniques, and the use of digital tools to support survivors effectively. Furthermore, they must be trained to differentiate between various causes of non-adherence—such as akrasia, cognitive challenges, physical limitations, or external pressures—to support targeted and appropriate interventions.

From a sociological perspective, achieving equitable access to personalized follow-up care across diverse CAYACS populations is a significant challenge. Systemic inequalities, such as socio-economic barriers or geographical disparities, exacerbate the issue of non-adherence, particularly in marginalized communities. Patient associations play a pivotal role in bridging these inequalities by providing advocacy, resources, and support networks that address gaps in care and amplify survivors’ voices in cancer pathways and care related to other conditions [122,123,124,125]. Ethically, this raises questions about justice in healthcare: how can we ensure that all survivors, regardless of their background, receive the tailored support they need to overcome these barriers? Addressing this care gap requires medical innovation and policy changes that address structural inequalities in healthcare.

Beyond these aspects, incorporating the perspectives of parents and caregivers into future ethical reflections may enrich our understanding of the relational and contextual factors shaping long-term follow-up behaviors. This line of inquiry is fully aligned with ongoing work within the e-QuoL project, which includes dedicated tasks focused on the needs of parents, siblings, and caregivers. The next phase of the project involves first the completion of a needs-assessment questionnaire, then the co-construction of tailored informational and support resources aimed at addressing the specific challenges faced by these groups.

### 4.2. Limitations

This article provides valuable insights into the concept of akrasia in childhood, adolescent, and young adult cancer survivors (CAYACS), but several limitations should be acknowledged. First, the concept of akrasia in this population remains underexplored in empirical research, with much of the analysis relying on theoretical models and extrapolations from broader behavioral science studies. Additionally, while the study emphasizes psychological and systemic barriers to adherence, it does not include longitudinal data to directly assess the efficacy of proposed interventions such as nudge theory or shared decision-making.

The diversity of CAYACS’ experiences and socio-cultural contexts further complicates the generalizability of the findings. Factors such as socioeconomic status, access to healthcare, and cultural attitudes toward medical follow-ups were not systematically analyzed, potentially limiting the applicability of recommendations across different regions or demographics.

Finally, the reliance on qualitative insights from an ethics reflection group provides depth but may lack the statistical rigor of quantitative studies. Future research should integrate mixed-methods approaches to evaluate the interplay of psychological, cognitive, and systemic factors in adherence behaviors, providing a more comprehensive understanding of akrasia in survivorship care.

## 5. Conclusions

Understanding the fundamental causes of nonadherence is essential for successful long-term follow-up programs for CAYACS. The concept of akrasia, though nuanced in this population, highlights how unique challenges like cognitive impairments, which compromise the freedom of decision, and psychological distress may obscure the notion of intentionality and complicate patients’ ability to act in their own best interest.

An important question emerges: is akrasia in CAYACS fundamentally different from akrasia in other populations? Trauma, disrupted developmental processes, and challenges related to social reintegration suggest it may manifest uniquely in this group. Exploring these dynamics further could pave the way for tailored interventions and deeper insights into survivors’ adherence behaviors.

Effective solutions must respect patient autonomy while addressing specific barriers. Nudge theory and shared decision-making offer complementary approaches: while nudges subtly guide patients toward health-promoting behaviors, shared decision-making ensures that care plans align with individual values, fostering ownership and long-term commitment. Digital tools offer promising pathways for enhancing engagement. Co-creating these tools with survivors, as in an ongoing aspect of the e-QuoL project, aligns interventions with survivors’ values and fosters a sense of ownership, increasing the likelihood of adherence.

Future research should investigate the interplay of psychological, cognitive, and systemic factors shaping adherence behaviors. By integrating personalized strategies with systemic reforms, such as policy support for multidisciplinary care and digital innovation, we can better address the complexities of CAYACS care. Understanding and addressing akrasia can ultimately enhance adherence, improve QoL, and ensure equitable long-term outcomes for this vulnerable population.

## Figures and Tables

**Figure 1 cancers-17-01310-f001:**
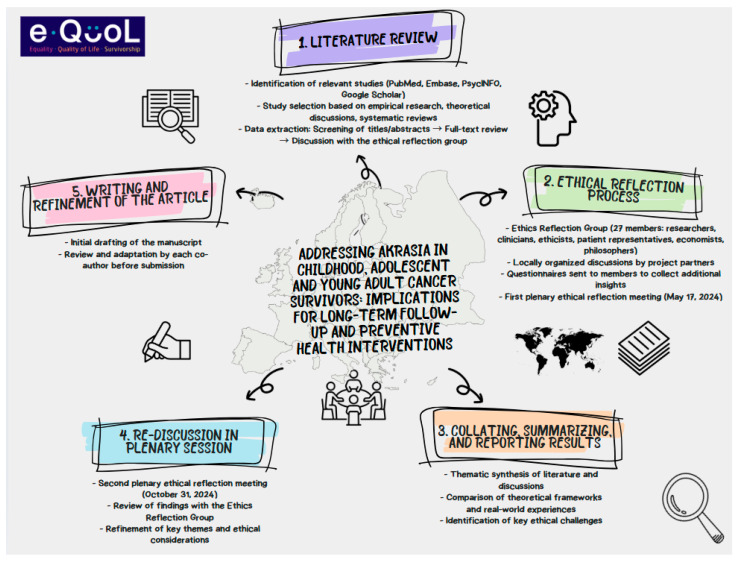
Methodological approach to ethical reflection in the e-QuoL project.

**Table 1 cancers-17-01310-t001:** Adherence challenges and interventions for CAYACS’ long-term follow-up care.

Challenges	Description	Proposed Interventions
Inconsistent Adherence	Survivors may skip follow-ups or have difficulties maintaining consistent engagement in healthy behaviors over time due to a lack of awareness or to psychological, economic, or logistical barriers. Adherence tends to fluctuate with life stages, changing priorities, and motivation.	Tailored communication, shared decision-making, and flexible care models.
Psychological Barriers	Anxiety, fear of recurrence, and post-traumatic stress disorder (PTSD) disrupt self-care and health engagement. Prioritization of short-term satisfaction over long-term health due to trauma and fatigue.	Psychological support and counseling (individual or group therapy; cognitive–behavioral therapy or other modalities), motivational interviewing, and peer support. Communication, information, education.
Cognitive Impairments	The cancer itself, as well as treatments like chemotherapy and radiotherapy can impair education, decision-making, and executive functions.	Simplified follow-up plans and digital tools with adaptive, interactive content. Neuropsychological assessment and cognitive training or cognitive rehabilitation therapy for cognitive dysfunction.
Economic and Logistical Challenges	Costs, travel, and scheduling conflicts hinder access, especially in underserved areas.	Financial support, telemedicine, and regional care adaptations.
Social Isolation and Reintegration	Survivors face disrupted schooling, employment, and peer relationships, reducing quality of life (QoL), capacity for decision-making, and autonomy.	Community support networks and school/work-reintegration programs.
Akrasia	A tendency to act against one’s better judgment.	Nudges (e.g., reminders), decision aids, and patient-activation programs. Welcoming and non-judgmental environment, simple access to care.

## Data Availability

Not applicable.

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
