# Peer review of "Addressing Akrasia in Childhood, Adolescent and Young Adult Cancer Survivors: Implications for Long-Term Follow-Up and Preventive Health Interventions"

_cancers, 2025, doi:10.3390/cancers17081310_

Round 1

Reviewer 1 Report

Comments and Suggestions for Authors

Thank you for the opportunity to review this manuscript. I believe this article will provide an important addition to the literature on a timely and vital topic. I have no concerns about the rigor due to the qualitative nature of the project design. I believe a couple of components should be considered for future work (or if they were included in the present work, should be elucidated more clearly). 1) in addition to the philosophical contributions, a spiritual dimension should be considered. Including a member of the committee who has a background with chaplaincy would offer an additional more comprehensive element to discussions of importance of long life, health, purpose, etc. when trying to comprehend  LTFU. If this is the element contributed by "philosophers" then perhaps explicit inclusion of that fact would be beneficial to the present manuscript. 
2) Authors discuss family members and caregivers in the discussion when talking about implications for follow-up care and potential policy implications. However, I also think these perspectives could be a valuable addition to future work - including the ethical reflections to provide context and additional information to the basic findings.

Author Response

Dear Reviewers,

We sincerely appreciate your thoughtful and constructive feedback on our manuscript. Your insights helped us refine our work and improve its clarity and impact. Below, we provide detailed responses (in blue) to each of your comments and outline the corresponding revisions made to the manuscript (in blue in the manuscript or with track changes).

We hope that our revisions address your concerns adequately and improve the overall quality of our manuscript.

Additionally, a new figure was added to the manuscript to present the method and the English was reviewed by our native English authors.

We have also reviewed the affiliations and corrected the bibliography to the MPDI rules. This last part is not with a tracked version as all was adapted.

Reviewer 1:

Thank you for the opportunity to review this manuscript. I believe this article will provide an important addition to the literature on a timely and vital topic. I have no concerns about the rigor due to the qualitative nature of the project design. I believe a couple of components should be considered for future work (or if they were included in the present work, should be elucidated more clearly).

1) in addition to the philosophical contributions, a spiritual dimension should be considered. Including a member of the committee who has a background with chaplaincy would offer an additional more comprehensive element to discussions of importance of long life, health, purpose, etc. when trying to comprehend LTFU. If this is the element contributed by "philosophers" then perhaps explicit inclusion of that fact would be beneficial to the present manuscript.

Response: We appreciate the suggestion to include a spiritual dimension in our discussion. While our current analysis primarily focuses on philosophical perspectives, we recognize that spiritual considerations could enrich the discourse on long-term follow-up (LTFU), health, and purpose. In response, we have clarified the role of philosophical contributions in addressing such existential concerns. Additionally, we have acknowledged the potential value of integrating chaplaincy perspectives in future research and interdisciplinary discussions. That said, given the diversity of spiritual and religious backgrounds among survivors, we believe that any such inclusion should be approached with attention to plurality and inclusivity, to avoid privileging a specific tradition.

In the limitation section we added “While our analysis emphasizes ethical and philosophical dimensions, we acknowledge that spiritual beliefs may also influence how survivors relate to health, illness, and long-term follow-up care. Future interdisciplinary work could consider the role of spir-itual frameworks—when approached with attention to religious diversity—as part of a broader understanding of survivor motivation and engagement.”

2) Authors discuss family members and caregivers in the discussion when talking about implications for follow-up care and potential policy implications. However, I also think these perspectives could be a valuable addition to future work - including the ethical reflections to provide context and additional information to the basic findings.

Response: We acknowledge the importance of including family members and caregivers in ethical reflections on long-term follow-up (LTFU). While our present study does not incorporate direct empirical accounts from these groups, we have revised the discussion to emphasize their potential contributions to future research. This direction is fully aligned with the broader objectives of the e-QuoL project, to which this ethics reflection group belongs. Within e-QuoL, a dedicated team is actively working on integrating caregiver and family perspectives, with specific deliverables planned on this topic. Furthermore, the views of parents have already been partially represented in this work, as some of the co-authors are members of parent advocacy associations, ensuring their voices inform the ethical considerations discussed in the manuscript.

We added this in the discussion “Beyond these aspects, incorporating the perspectives of parents and caregivers into future ethical reflections may enrich our understanding of the relational and contextual factors shaping long-term follow-up behaviors. This line of inquiry is fully aligned with ongoing work within the e-QuoL project, which includes dedicated tasks focused on the needs of parents, siblings, and caregivers. Following the completion of a needs-assessment questionnaire, the next phase of the project involves the co-construction of tailored informational and support resources aimed at addressing the specific challenges faced by these groups.”

Reviewer 2 Report

Comments and Suggestions for Authors

important topic 

However, some revisions are required.

In abstract and background, please clearly state the definition of akrasia upfront.

Methods: unclear whether this is review paper, secondary data analysis, or data collection as the first time study. clarify this in the abstract as well as main manuscript

how to recruit them and how many samples, duration, who provided the surveys etc.

authors mentions that arkasia is different compared to healthy control. We did not have clear literature to support this. please clarify this sentence by adding literature outhere. However, if no literature, delete this sentence. 

authors continuously mentioned that the framework

if there is specific framework, please add the framework as figure. it is hard to follow without figure and unclear what framework is mentioned. 

Comments on the Quality of English Language

moderate revision is required 

Author Response

Dear Reviewers,

We sincerely appreciate your thoughtful and constructive feedback on our manuscript. Your insights helped us refine our work and improve its clarity and impact. Below, we provide detailed responses (in blue) to each of your comments and outline the corresponding revisions made to the manuscript (in blue in the manuscript or with track changes).

We hope that our revisions address your concerns adequately and improve the overall quality of our manuscript.

Additionally, a new figure was added to the manuscript to present the method and the English was reviewed by our native English authors.

We have also reviewed the affiliations and corrected the bibliography to the MPDI rules. This last part is not with a tracked version as all was adapted.

Reviewer 2:

important topic

However, some revisions are required.

In abstract and background, please clearly state the definition of akrasia upfront.

Response: We have revised the abstract and background to clearly define akrasia upfront, ensuring that readers are provided with a precise understanding of the term before engaging with the rest of the manuscript. Akrasia is a difficult concept to understand. Thus, examples were provided in the background section after the definition.

Methods: unclear whether this is review paper, secondary data analysis, or data collection as the first time study. clarify this in the abstract as well as main manuscript.

How to recruit them and how many samples, duration, who provided the surveys etc.

Response: We acknowledge that our description of the methodology was unclear. We have now explicitly stated the methodology used: a literature review followed by an ethic reflection group. The analysis of the data was based on the discussion of the ethic reflection group. We added here a figure to explain the methods.

authors mentions that arkasia is different compared to healthy control. We did not have clear literature to support this. please clarify this sentence by adding literature outhere. However, if no literature, delete this sentence.

Response: We have deleted this sentence “The challenges at the end of active treatment and throughout survivorship are distinct for CAYACS compared to the general population.” As this does not mean anything as the general population does not need medical follow-up.

We changed by introducing this gap in knowledge while introducing the hypothesis “.Although no studies to date have directly compared akrasia in CAYACS to that in healthy populations, the psychosocial and medical challenges faced by survivors—including trauma, disrupted development, and complex follow-up trajectories—suggest a distinctive context in which akratic behaviors may manifest differently.”

authors continuously mentioned that the framework

if there is specific framework, please add the framework as figure. it is hard to follow without figure and unclear what framework is mentioned.

Response: We appreciate the reviewer’s comment regarding the repeated use of the term framework. Upon review, we agree that the word was used in different contexts—sometimes to refer to theoretical models, and other times to describe general approaches or perspectives (e.g., ethical, behavioral), or to describe the e-QuoL project. To improve clarity, we have revised the manuscript to limit the use of the term. As no single unified theoretical model was presented, but rather a combination of theoretical insights, we did not add a figure. Instead, we clarified the terminology to avoid confusion. We also refered to the e-QuoL website to explain how it is incorporated in this global project.